# Methylglyoxal, Glycated Albumin, PAF, and TNF-α: Possible Inflammatory and Metabolic Biomarkers for Management of Gestational Diabetes

**DOI:** 10.3390/nu12020479

**Published:** 2020-02-14

**Authors:** Gabriele Piuri, Katia Basello, Gabriele Rossi, Chiara Maria Soldavini, Silvia Duiella, Giulia Privitera, Angela Spadafranca, Andrea Costanzi, Emiliana Tognon, Mattia Cappelletti, Paola Antonia Corsetto, Angela Maria Rizzo, Attilio Francesco Speciani, Enrico Ferrazzi

**Affiliations:** 1Inflammation Society, 18 Woodlands Park, Bexley DA52EL, UK; gabriele.piuri@me.com (G.P.); m.cappelletti@me.com (M.C.); enrico.ferrazzi@unimi.it (E.F.); 2GEK lab–Cryolab, University of Rome Tor Vergata, Via Montpellier, 1—00133 Rome, Italy; katia.basello@gek-group.com (K.B.); andrea.costanzi@gek-group.com (A.C.); emiliana.tognon@gek-group.com (E.T.); 3Obstetrical Unit, Woman-Child-Newborn Department, Fondazione IRCCS Ca’ Granda, Ospedale Maggiore Policlinico, 20122 Milan, Italy; gabriele.rossi@policlinico.mi.it (G.R.); chiaramaria.soldavini@gmail.com (C.M.S.); silvia.duiella@policlinico.mi.it (S.D.); giulia.privitera@policlinico.mi.it (G.P.); angela.spadafranca@gmail.com (A.S.); 4Department of Pharmacological and Biomolecular Sciences, Università degli Studi di Milano, 20133 Milano, Italy; paola.corsetto@unimi.it (P.A.C.); angelamaria.rizzo@unimi.it (A.M.R.); 5Fondazione IRCCS Cà Granda, Ospedale Maggiore Policlinico, Department of Clinical Sciences and Community Health, Università degli Studi di Milano, 20122 Milan, Italy

**Keywords:** gestational diabetes, platelet-activating factor, tumor necrosis factor α, methylglyoxal, glycated albumin

## Abstract

Background: In gestational diabetes mellitus (GDM), pancreatic β-cell breakdown can result from a proinflammatory imbalance created by a sustained level of cytokines. In this study, we investigated the role of specific cytokines, such as B-cell activating factor (BAFF), tumor necrosis factor α (TNF-α), and platelet-activating factor (PAF), together with methylglyoxal (MGO) and glycated albumin (GA) in pregnant women affected by GDM. Methods: We enrolled 30 women whose inflammation and metabolic markers were measured at recruitment and after 12 weeks of strict dietetic therapy. We compared these data to the data obtained from 53 randomly selected healthy nonpregnant subjects without diabetes, hyperglycemia, or any condition that can affect glycemic metabolism. Results: In pregnant women affected by GDM, PAF levels increased from 26.3 (17.4–47.5) ng/mL to 40.1 (30.5–80.5) ng/mL (*p* < 0.001). Their TNF-α levels increased from 3.0 (2.8–3.5) pg/mL to 3.4 (3.1–5.8) pg/mL (*p* < 0.001). The levels of methylglyoxal were significantly higher in the women with GDM (*p* < 0.001), both at diagnosis and after 12 weeks (0.64 (0.46–0.90) μg/mL; 0.71 (0.47–0.93) μg/mL, respectively) compared to general population (0.25 (0.19–0.28) μg/mL). Levels of glycated albumin were significantly higher in women with GDM (*p* < 0.001) only after 12 weeks from diagnosis (1.51 (0.88–2.03) nmol/mL) compared to general population (0.95 (0.63–1.4) nmol/mL). Conclusion: These findings support the involvement of new inflammatory and metabolic biomarkers in the mechanisms related to GDM complications and prompt deeper exploration into the vicious cycle connecting inflammation, oxidative stress, and metabolic results.

## 1. Introduction

The definition of gestational diabetes (GDM) is related to high blood glucose levels that appear for the first time during gestation. In the latter stages of gestation, a pancreatic response to progressive insulin resistance occurs, and GDM is the failure of the pancreas to appropriately increase the β-cell mass and the connected insulin secretion. Around 4% to 10% of all pregnant women develop GDM [1,2]. The predisposition to developing GDM depends on different diagnostic criteria, including individual history of obesity and maternal age at first pregnancy. GDM is also associated with a highly and significantly increased risk of maternal and neonatal morbidity [2]. Among the long-term complications, there is an increased risk of developing type two diabetes in adult life, and GDM can negatively influence epigenetic programming in the fetus and newborn development, increasing their cardiovascular and metabolic risks [3].

Several reports have shown that the systemic inflammatory response is more pronounced in women with GDM [4]. Several inflammatory mediators may play an essential role in GDM pathogenesis. Therefore, knowing the levels of these mediators and their effects during gestation may create new opportunities for improving diagnosis and cures for GDM and preventing GDM complications. 

The placenta physiologically contributes to inflammation and insulin resistance by secreting proinflammatory cytokines. In GDM, pancreatic β-cell breakdown can result from the proinflammatory imbalance created by a sustained level of cytokines [5]. In this study, we investigated the role of specific cytokines, such as B-cell activating factor (BAFF), tumor necrosis factor α (TNF-α), and platelet-activating factor (PAF), together with some markers of oxidative stress, in pregnant women affected by GDM. It has already been observed that those signaling cytokines and inflammatory molecules have a role in the mechanisms of insulin resistance and may result in pregnancy complications such as maternal hypertension [6,7,8,9].

BAFF plays a role in determining insulin resistance [10,11,12] and has been implicated in GDM. BAFF is an essential immune regulator that was recently reported to be secreted by the placenta [6,13].

High levels of TNF-α reduce insulin sensitivity by disrupting the translocation of glucose transport GLUT-4 channels and insulin signal transduction [14]. The expression of TNF-α in the placenta is thought to contribute to the insulin resistance associated with pregnancy. However, this TNF-α increase is controversial [15] and is possibly linked to body mass index (BMI) before pregnancy [16].

PAF is another important mediator of inflammation, and its highly unstable molecules could contribute to the pathogenesis of pre-eclampsia [17]. However, no studies have investigated its possible role in GDM.

In GDM, hyperglycemia induces the overproduction of methylglyoxal (MGO), a circulating toxic intermediate metabolite that enters the fetal circulation and crosses cell membranes [18,19]. MGO is an oxidizing substance, and its concentration is directly correlated with blood sugar levels. MGO is controlled via an enzymatic system, which detoxifies the organism by converting 99% of the MGO into less reactive products. An excessive accumulation of MGO results in the addition of this molecule to proteins and DNA, leading to oxidative stress, cellular aging, DNA mutations, and pro-apoptotic effect. Krishnasam et al. also demonstrated that the concentrations of MGO are significantly higher in GDM women compared to MGO in normal pregnant women [20]. Chang and Chan found that MGO has harmful effects on early-stage oocyte maturation and fertilization [21]. Besides its direct effects, MGO is a precursor to advanced glycation end-products (AGEs), which, in turn, are teratogenic. AGEs are formed either directly via a nonenzymatic reaction between glucose and the N-terminal part of proteins, or indirectly through α-oxoaldehydes. 

Approximately 60–70% of total serum proteins are represented by human serum albumin, which is the most abundant extracellular protein in plasma. Human serum albumin is a multifunctional protein, and interest has increased in its activity as a biomarker of hyperglycemia due to its high sensitivity to glycation. Several differences exist between HbA1c and albumin, and the latter has a rate of nonenzymatic glycation that is approximately 10-fold higher than that of hemoglobin [22]. The higher susceptibility to glycation of albumin and other plasma proteins compared to intracellular proteins like hemoglobin induces some specific effects. The blood levels of glycated albumin exhibit a broader fluctuation than those of HbA1c, allowing earlier detection of rapid changes in blood glucose [23,24]. Few studies have investigated the association between glycated albumin in diabetic pregnant women, and the results about the complications in their children have been conflicting [25,26,27].

This study aimed at investigating the longitudinal trends of these inflammatory and metabolic molecules during pregnancy to demonstrate new pathogenetic markers of GDM. The secondary aim of this study was evaluating the possible correlation between these data to routine metabolic analyses and anthropometric measurements of mothers and newborns.

## 2. Materials and Methods 

### 2.1. Study Design and Subjects

We conducted this observational study on a cohort of Caucasian pregnant women with GDM, recruited at the Department of Obstetrics and Gynecology, Fondazione IRCCS Ca’ Granda, Ospedale Maggiore Policlinico, Milan, Italy. A two-hour, 75-g oral glucose tolerance test (OGTT), performed at 24–28 weeks according to the International Association of Diabetes and Pregnancy Study Groups (IADPSG) criteria [28], was used to confirm the diagnosis of GDM. Exclusion criteria were chronic gastrointestinal diseases; pre-pregnancy diabetes; celiac disease; a history of eating disorders, such as anorexia or bulimia; vegan, vegetarian, or macrobiotic regimens; and non-Caucasian ethnicity. The institutional board reviewed and approved the study procedures and each subject provided written informed consent. We conducted the study following the Declaration of Helsinki. The local ethical committee approved the study protocol (Clinical Trial Center Fondazione IRCCS Ca’ Granda Ospedale Maggiore Policlinico, project identification code 4004, approval number 126_2018 on 28 March 2018). For the general population, we present data obtained from 53 randomly selected healthy nonpregnant subjects, matched to GDM women for age, without diabetes, hyperglycemia, or any condition that can affect glycemic metabolism. The data of this group were derived from an internal database related to healthy nonpregnant subjects who released informed consent for scientific purpose. We collected samples using a finger prick and analyzed the total albumin, GA, and MGO.

We recorded anthropometric measures and fasting blood sample collection to measure serum inflammatory markers (BAFF, PAF, TNF-α, MGO, and GA) during the recruitment visit three days after the OGTT and 12 weeks after GDM diagnosis. All subjects had to present complete medical histories and undergo a physical examination, anthropometric assessment and routine laboratory tests including fasting blood glucose, postprandial blood glucose, glycated hemoglobin, insulin, cortisol at 08:00, total cholesterol, HDL cholesterol, LDL cholesterol, triglycerides, CRP, creatinine, and ferritin. Maternal venous blood was obtained from peripheral venipuncture, and the serum was stored at −80 °C. In this observational study, all OGTT were performed in the same analysis laboratory (Fondazione IRCCS Ca’ Granda, Ospedale Maggiore Policlinico, Milan, Italy). All the women whose OGTT levels were positive for a diagnosis of GDM were called by the physician to invite them to participate in this observational study.

### 2.2. Anthropometry

The same operator recorded the anthropometric measurements at the recruitment visit according to standard criteria and measuring procedures [29]. The pregnant women wore only underwear. The operator measured their weight (to the nearest 0.1 kg) and their standing height (SH; to the nearest 0.1 cm) using the same calibrated scale, which had a telescopic vertical steel stadiometer (SECA 711, seca Deutschland, Hamburg, Germany). We calculated the BMI as weight (kg)/stature^2^ (m^2^). We also calculated pre-pregnancy BMI using the self-reported body weight before pregnancy. Biceps, triceps, and subscapular skinfolds were measured using a Tanner–Whitehouse caliper (Holtain Ltd., Crosswell, U.K.), measuring each skinfold three times and using the mean value for analysis.

### 2.3. Dietary Intervention

All women received a dietary program to control their glucose metabolism and weight gain. If necessary, their diets were hypocaloric to provide at least the resting energy expenditure estimated by the Harris–Benedict formula considering the pre-pregnant weight and energy surplus linked to gestational age following the Recommended Assumption Levels of Energy and Nutrients for the Italian Population (LARN) [30].

The proposed nutritional pattern agreed with the Healthy Eating Plate proposed by Harvard School of Public Health [31]. The daily total energy intake was distributed across three main meals (breakfast, lunch, and dinner) and two snacks.

The macronutrient composition was balanced as follows: 45% of the total energy from carbohydrates, of which simple sugars comprised less than 12%; 25–35% of total energy from fat, of which less than 7% was from saturated fat and 10% from polyunsaturated fatty acids (PUFAs). The protein intake satisfied the pregnancy requirements, as indicated in the LARN, with 50% derived from vegetable sources and 50% from animal sources. The quality of protein intake was regulated by the following frequencies of consumption: Meat, preferably white, 2 times/week; fish 2–3 times/week, with a preference for blue fish for optimal intake of omega-3 fatty acids; legumes 3–4 times/week; eggs 2 times/week, cheese 1–2 times/week; ham 1 time/week; and nuts 20–30 g every day. Food with a high glycemic index were prohibited. Two servings of fruit and three servings of vegetables were advised to be consumed daily. Olive oil was indicated as the main culinary lipid. Dietary cholesterol was lower than 200 mg/day and fiber intake was about 30 g, in agreement with the Guidelines for Healthy Nutrition.

In order to assess the adherence to the diet and the effect on glycemic profile, every woman was invited to fill a daily food diary and to monitor their glycemic profile twice a day via the use of a glucometer. Fasting glycemic cutoff was 95 mg/dL, while 2h postprandial glucose cutoff was 120 mg/dL. Every two weeks, a meeting with the dietitian and the gynecologist was proposed to evaluate the health status of the fetus and maternal body weight and fat mass. Clinical dietitians were responsible for providing dietary counseling for GDM women.

Subjects included in the control group did not receive any dietary counseling.

### 2.4. Enzyme-Linked Immunosorbent Assay (ELISA)

Human glycated albumin (GA) and total albumin (TA) concentrations in the plasma samples were determined using a Human Glycated Albumin ELISA Kit (lower range of detection 19.53 pmol/mL, sensitivity <11.719 pmol/mL, Catalog No. abx252493, Abbexa Ltd., Cambridge Science Park, Cambridge, U.K.) and a Human Albumin Immunoperoxidase Assay for the Determination of Albumin kit (lower range of detection 0 ng/ml, Human Samples, Catalog No. E-80AL, Immunology Consultants Laboratory, Portland, OR, USA), respectively. The levels of GA and serum albumin are expressed as a percentage to exclude the influence of serum albumin. We calculated the GA% using the following formula: GA% = GA (μmol/mL)/Total Albumin (μmol/mL) × 100.

MGO was measured using the OxiSelect™ Methylglyoxal Competitive ELISA Kit (lower range of detection 0 μg/mL, Catalog No. STA-811, Cell Biolabs, San Diego, CA, USA), which is an enzyme immunoassay developed for the detection and quantization of protein adducts of methylglyoxal-hydro-imidazoline (MG-H1).

The serum TNF-α, B-cell Activating factor (BAFF), and Human Platelet Activating Factor (PAF) were measured via commercial ELISA kits (Human TNF-a Ultrasensitive ELISA Kit, lower range of detection 0 pg/mL, sensitivity < 0.09 pg/mL, Catalog Number KHC3014, Invitrogen, ThermoFisher Scientific, Carlsbad, CA, USA; Human BAFF/BLyS/TNFSF13B Immunoassay Quantikine® ELISA, lower range of detection 0 pg/mL sensitivity 2.68 pg/mL, Catalog Number PDBLYS0B, R&D Systems Inc, Minneapolis, MN, USA; Human Platelet Activating Factor ELISA Kit, lower range of detection 0.313 ng/mL, sensitivity 0.188 ng/mL, Catalog Number E-EL-H2199, Elabscience Houston, TX, USA, respectively) using the Biomek 4000 ELISA microplate liquid reagent dispensing automation tool (Beckman Coulter, Brea, CA, USA) and the EL405LS ELISA microplate automated washing system (BioTek Instruments, Winooski, VT, USA). The absorbance of each well was read at a wavelength of 450 nm with a Multiskan FC plate reader (Thermo Scientific, Waltham, MA, USA). The average zero standard optical density was subtracted from all absorbances, and a standard curve was generated using a four-parameter logistic (4-PL) curve fit. The concentration in the test sample was calculated through interpolation along the standard curve by multiplying the result by the dilution factor.

### 2.5. Statistical Analysis

Statistical analysis of the data was performed using GraphPad Prism 8 for macOS (GraphPad Software, San Diego, CA, USA. Version 8.3.0 (328), 16 October 2019). The median and interquartile range (IQR) were calculated for each variable. The medians were compared using the Mann–Whitney test. The linear correlations were studied using Spearman’s rank correlation coefficient. A *p*-value < 0.05 was used as the limit of statistical significance.

## 3. Results

We enrolled 30 women whose mean maternal age at recruitment was 34.0 (32.7–38.5) years. Their mean gestational age at recruitment with a positive OGTT was 26 weeks ±6 days (25+4 – 27+4). Table 1 and Table 2 provide the anthropometric and metabolic data of the enrolled patients at recruitment and after 12 weeks of dietetic treatment, respectively. Body weight and waist circumference significantly increased as expected, while fat mass, measured by skinfolds, reduced. The median fasting and postprandial plasma glucose levels were within normal ranges both at T0 and after 12 weeks. Table 1 provides more details about delivery and newborns (53% females). Two newborns had a birth weight percentile higher than the 90th. The vast majority of the enrolled population did not experience any complications during delivery and in postpartum except for a case of third-degree laceration and a case of diastasis episiorrhaphy.

Inflammation markers were measured at recruitment and after 12 weeks of strict dietetic therapy. PAF levels increased from 26.3 (17.4–47.5) ng/mL to 40.1 (30.5–80.5) ng/mL (*p* < 0.001). TNF-α levels increased from 3.0 (2.8–3.5) pg/mL to 3.4 (3.1–5.8) pg/mL (*p* < 0.001). No significant changes were observed in the levels of BAFF, methylglyoxal, and glycated albumin, and in the ratio of glycated albumin to total albumin (Figure 1).

In the general population (age: 36.2 (31.1–39.3) years), the fasting blood glucose was 85.1 (80.3–92.5) mg/dL, and glycated hemoglobin was 33.1 (29.9–34.1) mmol/mol (as median (IQR), *p* ns compared to GDM group). As shown in Table 3, the levels of MGO were significantly higher in women with GDM (*p* < 0.001), both at diagnosis and after 12 weeks (0.64 (0.46–0.90) μg/mL; 0.71 (0.47–0.93) μg/mL, respectively) compared with general population at 0.25 (0.19–0.28) μg/mL. Levels of glycated albumin were significantly higher in women with GDM (*p* < 0.001) only after 12 weeks of diagnosis (1.51 (0.88–2.03) nmol/mL) compared with general population (0.95 (0.63–1.4) nmol/mL). We found no differences in the ratio of glycated albumin to total albumin.

### Correlations

PAF levels at diagnosis and after 12 weeks were positively correlated with glycated hemoglobin levels (*r* = 0.394, *p* = 0.031; *r* = 0.364, *p* = 0.048, respectively) and the HOMA index (*r* = 0.479, *p* = 0.018 and *r* = 0.422, *p* = 0.040, respectively).

We observed a positive correlation between MGO levels and HbA1c both at diagnosis (*r* = 0.401, *p* = 0.028) and after 12 weeks (*r* = 0.458, *p* = 0.011). MGO was significantly correlated with the HOMA index only at diagnosis (*r* = 0.440, *p* = 0. 031). MGO levels were positively correlated with both the pre-pregnancy weight (*r* = 0.400, *p* = 0.028) at GDM diagnosis and after 12 weeks (*r* = 0.406, *p* = 0.026), and with the birth weight (*r* = 0.438, *p* = 0.032).

Glycated albumin levels at GDM diagnosis were positively correlated with pre-pregnancy weight and BMI (*r* = 0.388, *p* = 0.037 and *r* = 0.417, *p* = 0.024, respectively).

## 4. Discussion

After the enrollment of pregnant women diagnosed with positive OGTT at a gestational age of around 26 weeks, we observed the following:
(1)After the diagnosis, the metabolic parameters of the women following the suggested diet substantially stabilized. The median values of fasting and postprandial glucose levels were normal both at T0 and after 12 weeks, while HbA1c showed a modest increase, but their values remained within the normal range. The same was observed for MGO and GA, which both showed a slight increase in their values but without any statistical significance. The only significant increase was observed in the triglycerides values, confirming the results already shown in literature [32,33]. Despite the stability of the standard glycemic parameters, this increase could reflect an obesogenic pathway [34] due to an inflammatory and metabolic effect on insulin resistance of PAF, TNF-alfa [35], MGO [36], and GA [37], that needs further investigation.(2)Maternal body weight increased after 12 weeks, while fat mass reduced. This result, also associated with a correct evaluation of the food diaries, suggests an excellent dietetic adherence and a positive effect of the dietary intervention on body composition.(3)Despite this metabolic stability, a significant increase of two inflammatory cytokines (PAF and TNF-α) was observed, corresponding to the proinflammatory conditions of gestational diabetes mellitus (GDM), acting even without metabolic impairment.(4)Despite normal HbA1c and fasting glycaemia levels, the metabolic biomarkers MGO and GA were significantly different compared with the general population. For MGO, this difference was evident since the time of diagnosis at around 26 weeks.(5)Some positive correlations were observed among inflammatory markers, metabolic parameters, and the anthropometric analysis. For example, a strict correlation between MGO and fetal overgrowth was evident, and a correlation between PAF, MGO, and the HOMA index was also observed.

Overall, this evidence indicates the possibility of using these biomarkers to better understand how to prevent possible future GDM complications for mothers and their children.

Pregnancy is one of the most rapidly stressful metabolic events in the life of a woman. Women with a genetic and metabolic predisposition to GDM, even at a young age, can develop GDM, which lasts only during pregnancy due to its metabolic burden. The appearance of this disease is often a prediction of future development of type 2 diabetes, for which age, weight, and other metabolic factors can determine similar conditions of stress, metabolic burden, and oxidative or inflammatory overload.

Until present, the diagnosis of GDM has been based on the OGTT, which is useful for identification of the disease but cannot be used to predict future complications or precise phenotypic correlations like BMI before pregnancy and fetal overgrowth. The use of markers like GA may not be suitable for the diagnosis and the management of GDM due to the reduced duration of episodes of hyperglycemia. According to a longitudinal study of the concentration of MGO and GA, we hypothesize that MGO could be useful for the identification of women with a high risk of GDM and, with the support of OGTT, the diagnosis and prediction of complications (in particular for the prediction of fetal overgrowth). However, GA may not be useful for an accurate diagnosis, but could be used as an excellent metabolic biomarker to evaluate glycemic control in GDM women, similar to how HbA1c is now used in diabetic patients. The identification of new markers for metabolic distress in GDM could lead to a better follow-up for GDM women.

The development of GDM could also be induced by an increase in the inflammatory conditions connected to the TNF-α and PAF pathways, to oxidative stress due to the glycation of many proteins, or due to the production of oxidative substances like MGO. To the best of our knowledge, this is the first time in which inflammatory biomarkers like PAF and TNF-α have been studied longitudinally during GDM pregnancies. The increase in PAF and TNF-α 12 weeks after the diagnosis of GDM could be useful in the future for monitoring this condition and potentially help to explain the hidden mechanisms behind metabolic and inflammatory interactions in GDM.

### Limitations

The population analyzed was too small to define precise future guidelines, and further studies with a larger cohort are needed for a better definition. The general population used for the comparison of metabolic values was nonidentical to the GDM-affected women in this study. In future studies, healthy pregnant women should be compared with those who are affected by GDM. However, according to Krishnasam [20], the concentrations of MGO are already known to be significantly higher in GDM women compared with the MGO values in normal pregnant women. In that study, the reported absolute values differed compared to ours, although this may be ascribed to differences in the method, as Krishnaman et al. employed a diverse commercial ELISA methodology [20]. Our study population was represented only by Caucasian women due to the usual referrals to the hospital involved.

## 5. Conclusions

The aim of this study was to investigate the longitudinal trends of these inflammatory and metabolic molecules to demonstrate new pathogenetic markers of GDM. A specific diet in women with GDM is a crucial factor for the correct management of GDM itself. However, the increase of specific inflammatory biomarkers could exert a role that is potentially related to metabolic impairment. 

This study supports the involvement of new inflammatory and metabolic biomarkers in the mechanisms related to the complications of GDM and prompts a deeper exploration into the vicious cycle connecting inflammation, oxidative stress, and metabolic results.

## Figures and Tables

**Figure 1 nutrients-12-00479-f001:**
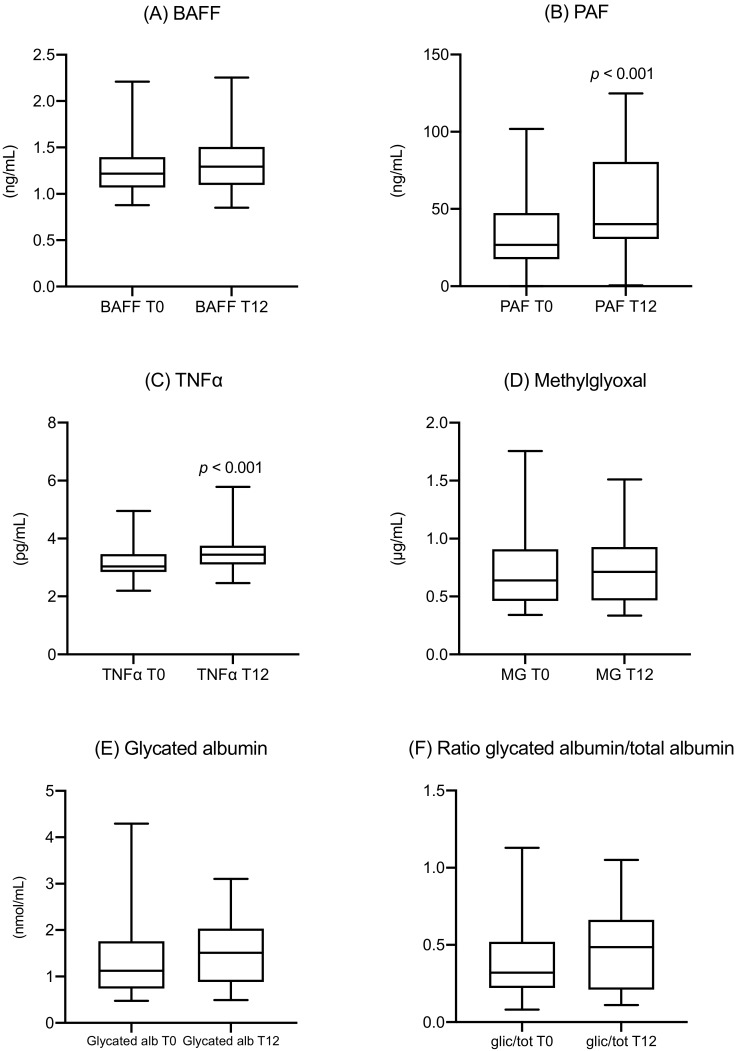
Longitudinal trend of inflammatory and metabolic markers in women with GDM at the time of diagnosis (T0) and after 12 weeks (T12): (**A**) BAFF, (**B**) PAF, (**C**) TNF-α, (**D**) methylglyoxal, (**E**) glycated albumin, (**F**) ratio of glycated albumin to total albumin. BAFF, B-cell activating factor; PAF, platelet-activating factor; TNF-α, tumor necrosis factor α; MG, methylglyoxal; GA, glycated albumin; glic/tot, ratio of glycated albumin to total albumin.

**Table 1 nutrients-12-00479-t001:** Anthropometric data of the enrolled women before and during pregnancy and details related to delivery and newborns.

Anthropometric Data	Before Pregnancy		*p*
**Height (cm)**	163 (160–168)		-
Pre-gestational weight (kg)	62.8 (55.6–69.6)		-
Pre-gestational BMI (kg/m^2^)	23.3 (21.0–26.3)		-
	**At diagnosis**	**After 12 weeks of diet**	
Weight (kg)	71.0 (63.5–78.5)	78.0 (64.6–82.8)	<0.001
Arm circumference (cm)	29.0 (26.9–30.1)	28.8 (28.0–31.3)	ns
Wrist circumference (cm)	15.0 (14.3–16.0)	15 (14.3–16.0)	ns
Waist circumference (cm)	96.0 (87.5–100.0)	104.0 (97.9–107.1)	<0.001
Bicipital skinfold (mm)	9.0 (7.8–13.3)	10.7 (7.2–12.8)	0.05
Tricipital skinfold (mm)	21.6 (18.0–28.7)	20.1 (16.8–25.8)	0.001
Subscapular skinfold (mm)	18.40 (13.40–25.20)	14.4 (12.3–24.0)	0.02
	**Delivery and newborn details**		
Gestational age at birth (weeks+days)	39+5 (39+0–39+6)		-
Birth weight (g)	3170 (3040–3460)		-
Birth weight centile	41.5 (22.5–67.8)		-
APGAR 1’	9 (9–9)		-
APGAR 5’	10 (10–10)		-

We measured weight, circumferences, and skinfolds at diagnosis of gestational diabetes mellitus (GDM) and after 12 weeks of diet. Data are expressed as the median and interquartile range.

**Table 2 nutrients-12-00479-t002:** Metabolic data of enrolled women at diagnosis of gestational diabetes mellitus and after 12 weeks of diet.

Metabolic Data	At Diagnosis	After 12 Weeks of Diet	*p*
Fasting blood glucose (mg/dL)	85.4 (79.4–90.8)	80.0 (73.0–90.0)	ns
Post prandial blood glucose (mg/dL)	94.4 (88.4–103.9)	97.1 (92.7–100.7)	ns
Glycated hemoglobin (mmol/mol)	30.5 (28.8–32.0)	33.0 (31.8–35.3)	<0.001
Insulin (μU/mL)	9.3 (5.5–14.3)	9.7 (7.4–15.3)	ns
HOMA index	1.54 (0.88–2.31)	1.45 (0.70–2.30)	ns
Cortisol at 08:00 (μg/dL)	27.6 (21.1–30.9)	27.0 (23.0–32.4)	ns
Total cholesterol (mg/dL)	258 (221–279)	267 (232–301)	ns
HDL cholesterol (mg/dL)	79 (65–87)	75 (66–87)	ns
LDL cholesterol (mg/dL)	142 (109–167)	146 (114–164)	ns
Triglycerides (mg/dL)	185 (150–208)	227 (221–282)	<0.001
CRP (mg/dL)	0.42 (0.17–0.63)	0.35 (0.23–0.60)	ns
Creatinine (mg/dl)	0.47 (0.42–0.62)	0.56 (0.50–0.67)	ns
Ferritin (ng/ml)	18 (12–35)	23 (16–32)	ns

Data are expressed as the median and interquartile range. HOMA index, homeostatic model assessment index; HDL cholesterol, high-density lipoprotein cholesterol; LDL cholesterol, low-density lipoprotein cholesterol; CRP, C-reactive protein.

**Table 3 nutrients-12-00479-t003:** Comparison between MGO and GA in general population and women with GDM at the diagnosis and after 12 weeks of diet.

		GDM Women		GDM Women	
	**General Population**	**At Diagnosis**	***p***	**After 12 Weeks of Diet**	***p***
MGO (μg/mL)	0.25 (0.19–0.28)	0.64 (0.46–0.90)	<0.001	0.71 (0.47–0.93)	<0.001
GA (nmol/mL)	0.95 (0.63–1.4)	1.12 (0.74–1.76)	ns	1.51 (0.88–2.03)	<0.001

Data are expressed as the median and interquartile range. MG, methylglyoxal; GA, glycated albumin.

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
