# Peer review of "Methylglyoxal, Glycated Albumin, PAF, and TNF-α: Possible Inflammatory and Metabolic Biomarkers for Management of Gestational Diabetes"

_nutrients, 2020, doi:10.3390/nu12020479_

Round 1

Reviewer 1 Report

The article is quite interesting and is investigating a novel research field on the link between inflammation and gestational diabetes. I have some comments to improve the article. Specific points:

The data have only a preliminary relevance. The authors enrolled a very low number of pregnants. I think that the number should be increased to produce conclusive findings. The rationale for the selection of variables/biomarkers assessed is unclear. Please improve this point in the introduction.  Lower range limits of detection for ELISA kits have to be specified. English language has to be improved.

Author Response

Dear Reviewer,

Thank you so much for your suggestions that improve the quality of our work. We would like to respond to your comments point by point.

1) This study is absolutely a preliminary work. We know that one of the most significant limits of our work is the number of patients enrolled. We have talked about this limitation in the discussion. We hope in the near future to increase the number of subjects to confirm our results.

2) In light of your suggestion, we have added more details about the rationale for the selection of the biomarkers assessed in the introduction.

3) We have also added the lower range of detection for ELISA kits in the section Material and Methods.

4) This article had already been revised for the English language by the MDPI editing service before our submission.

We have found a typographical error in the unit of measure for PAF, and we have corrected it.

We are looking for any other comments and suggestions hoping that this revision has improved the quality of this manuscript and might fit the vision and prestige of this special issue.

Gabriele Piuri, Attilio Speciani, and Enrico Ferrazzi

Reviewer 2 Report

This observational study of women with GDM describes changes in some novel biomarkers from diagnosis to 12 weeks post prescribed GDM nutrition counseling and clinical monitoring. Some interesting data is reported but the statistical analysis is suboptimal; the data could be explored in greater depth with more advance modeling. The significant change in plasma triglycerides, despite glycemic control, is not interrogated but could provide useful insight to the inflammatory pathways involved in GDM.

The study has several significant limitations, notably the absence of a proper control group and the limited measures obtained from the “comparison group” here may be compromised by the availability of only peripheral blood, but this is not discussed.

Detail on dietary adherence and birth size parameters beyond raw birth weight are needed to strengthen the validity of some conclusions drawn.

Abstract:

Line 18: Gestational Diabetes Mellitus (GDM) should be used as the full correct term for this condition throughout the manuscript.

Line 22-23: not clear from the aims and methods if there is a comparison non-GDM group in this study, yet results are presented as such.

Introduction:

Line 42-43: this sentence does not make sense – what is meant by “individual epidemiology” and how is that related to diagnostic criteria?

Line 46: “non-infective mother-to-child transmissible disease” – this is a misleading description of the nature of GDM, as the condition itself is not passed to the child but the condition may influence fetal development, which may increase offspring susceptibility to later obesity and cardiometabolic disease risk. Suggest re-phrasing this sentence to reference fetal programming mechanisms associated with GDM.

Line 90: what is meant by “longitudinal trends”? The objective should specify that these biomarkers are only being assessed in pregnancy

Methods:

Line 94-96: Was the comparison non-GDM group formally enrolled and consented to this study also? Since samples were obtained from these women, they would be considered participants.

Line 100: why were women following vegan, vegetarian, or macrobiotic regimens excluded?

Line 107-108: please explain why only peripheral blood was collected from the comparison group and justify if measures of albumin, GA and MGO are comparable from peripheral versus fasting venous blood collected from GDM women.

Line 139: correct “vegetal” to “vegetable”

Line 127-131: please specify if clinical dietitians were responsible for providing dietary counselling for GDM women and whether they were involved in the research study. Did any women from the comparison group receive dietary counselling for any other reasons not related to GDM?

Line 147-149: The prescribed nutrition plan appears to be very detailed but are there any dietary intake data available to determine adherence? It is possible that inadequate dietary improvements in some women could explain variation in the various metabolic/inflammatory markers measured after 12 weeks.

Detailed metabolic data are presented in the results (table 2) but not mentioned as part of the study aims or described in the methods section.

Lines 176-178: statistics section requires more detail, e.g. in which groups and for which variables were the Mann Whitney U and Pearson correlations performed? How about consideration for covariates such as maternal age and BMI? linear regression adjusting for BMI would be informative to test if the changes or association between some of the metabolic and inflammatory markers is independent of weight status. Also, for those inflammatory markers that change significantly in GDM women, anthropometric or metabolic measures could be examined as predictors of the post-12 week values adjusting for baseline (i.e. predictors of the magnitude of change).

Results:

Descriptives of the 53 non-GDM women should be presented also.

Tables 1 and 2: additional detail in the titles should be moved to footnotes.

Table 1: Suggest adding birthweight data also. Are birthweight centile data available or could they be computed to examine incidence of large-for-gestational age and its association with the metabolic/inflammatory biomarkers in this cohort?

Lines 205-210: would be helpful to see these data for GDM and non-GDM women in a table.

Lines 212-221: It is difficult to decipher from the text if the presented correlations relate to the baseline or post-12 week measures for each variable – suggest presenting a table of correlation coefficients for all biomarker variables at each time point. Would also be interesting to see correlations with other markers not mentioned here, such as triglycerides.

Discussion:

Line 238: fetal overgrowth was not studied, only raw birthweight but even that data is not described. LGA incidence should be added to the data if possible.

Line 250-241: GDM complications were not assessed in this study, so it is incorrect to suggest that the results and these novel biomarkers could help better prevent GDM and its complications.

Author Response

Dear Reviewer,

Thank you so much for your meticulous and thorough work, which has allowed us to improve the quality of our article. We have followed your suggestions in revising our work. We would like to respond to your comments point by point.

1) This work wants to show preliminary data about new biomarkers to better understand the pathogenetic mechanisms of GDM. The decision to present only a descriptive statistical analysis was dictated by the low number of the subjects examined. The application of more advanced mathematical models requires a larger sample size to be employed correctly.

2) Thanks for stimulating us to discuss the significant change in plasma triglycerides. We have added a comment about this question in the discussion.

3) We have already examined the limits of our work in the discussion. About the absence of a proper control group, we have discussed our results according to Krishnasam et al. We have added more details about the general population hoping that is enough to better describe the strengths and limits of this work.

4) We have added more details on dietary adherence and data about birth size and delivery.

5) We have excluded from this work those women following vegan, vegetarian, or macrobiotic regimens because these kinds of diet can influence, positively or negatively, glycemic metabolism.

6) The data of the general population group came from an internal database related to healthy non-pregnant subjects who released informed consents for scientific purpose. We collected samples using a finger prick and analyzed the total albumin, GA, and MGO. The levels of these molecules in capillary blood are proportional to those found in venous blood.

7) We have added a description of the general population, birthweight, birthweight centile, data about delivery and complications, and a table with the comparison between MGO and GA in general population and women with GDM at the diagnosis and after 12 weeks of diet. About correlations, we have provided a table with correlation coefficients and “p" for all biomarker variables at each time point. We think these data deserve to be included in an appendix to the article.

We have found a typographycal error in the unit of measure for PAF, and we have corrected it.

We are looking for any other comments and suggestions hoping that this revision has improved the quality of this manuscript and might fit the vision and prestige of this special issue.

Gabriele Piuri, Attilio Speciani, and Enrico Ferrazzi

Round 2

Reviewer 1 Report

No further comments.

Author Response

Dear Reviewer,

Thank you so much for your kind and precious work that improves the quality of our manuscript.

Gabriele Piuri, Attilio Speciani, and Enrico Ferrazzi

Reviewer 2 Report

The revised manuscript is generally improved in terms of data presentation and methods description. I have some further minor comments:

Abstract: The first sentence of the results section should clearly state the data relates to the GDM women only. Line 106-107: it is not clear what is meant by the secondary aim - compare to whom? "anthropometric measurements" would be more correct than "evaluations". Line 176: replace "suggestion" with "counseling" or "advice". Line 248-249: what is meant by "matched for age and glycemic parameters"? It is possible the GDM women were matched to non-GDM women by age, but this should be specified in the methods section. It cannot be possible they were matched for glycemic parameters as they are comparing these values between the populations. Also, please specify what the values in the text represent here - median (IQR) perhaps? Line 280-282: discussion around the raised triglycerides and possible mechanisms through inflammatory/glycemic pathways could be elaborated further with a discussion of existing literature around the topic.

Author Response

Dear Reviewer,

Thank you so much for your further suggestions that have allowed us to improve the quality of our article. We would like to respond to your comments point by point.

1) We have explained the first sentence of the Results section in the abstract and the sentence about the secondary aim of the study.

2) For glycemic parameters, we mean fasting blood glucose and glycated hemoglobin. We have rewritten this sentence to avoid any misunderstanding. We have added the concept of matched for age in the Materials and Methods section.

3) We have further elaborated the part of the discussion related to the increase of triglycerides, and we have discussed our hypothesis according to the existing literature.

Gabriele Piuri, Attilio Speciani, and Enrico Ferrazzi